# Carbon Dot Synthesis in CYTOP Optical Fiber Using IR Femtosecond Laser Direct Writing and Its Luminescence Properties

**DOI:** 10.3390/nano14110941

**Published:** 2024-05-27

**Authors:** Ruyue Que, Jean-Frédéric Audibert, Enrique Garcia-Caurel, Olivier Plantevin, Kyriacos Kalli, Matthieu Lancry, Bertrand Poumellec, Robert B. Pansu

**Affiliations:** 1CNRS, ENS Paris-Saclay, CentraleSupélec, LuMIn, Université Paris-Saclay, 91190 Gif-sur-Yvette, France; queruyue@hotmail.com (R.Q.); jaudiber@ppsm.ens-cachan.fr (J.-F.A.); robert.pansu@ens-paris-saclay.fr (R.B.P.); 2Institut Polytechnique de Paris, CNRS, École Polytechnique, LPICM, 91120 Palaiseau, France; enrique.garcia-caurel@polytechnique.edu; 3Laboratoire de Physique des Solides, CNRS, Université Paris-Saclay, 91405 Orsay, France; olivier.plantevin@universite-paris-saclay.fr; 4Nanophotonics Research Laboratory, Cyprus University of Technology, 3036 Limassol, Cyprus; 5Institut de Chimie Moléculaire et des Matériaux d’Orsay, CNRS, Université Paris-Saclay, 91405 Orsay, France; matthieu.lancry@u-psud.fr

**Keywords:** CYTOP, carbon dots, femtosecond laser, POF, photoluminescence, optical fiber sensor

## Abstract

Luminescent carbon dots (CDs) were locally synthesized in the core of CYTOP fibers using IR femtosecond laser direct writing (FLDW), a one-step simple method serving as a post-treatment of the pristine fiber. This approach enables the creation of several types of modifications such as ellipsoid voids. The CDs and photoluminescence (PL) distribute at the periphery of the voids. The PL spectral properties were studied through the excitation/emission matrix in the visible range and excitation/emission spectra in the UV/visible range. Our findings reveal the presence of at least three distinct luminescent species, facilitating a broad excitation range extending from UV to green, and light emission spanning from blue to red. The average laser power and dose influence the quantity and ratio of these luminescent CD species. Additionally, we measured the spatially resolved lifetime of the luminescence during and after the irradiation. We found longer lifetimes at the periphery of the laser-induced modified regions and shorter ones closer to the center, with a dominant lifetime ~2 ns. Notably, unlike many other luminophores, these laser-induced CDs are insensitive to oxygen, enhancing their potential for display or data storage applications.

## 1. Introduction

Carbon dots (CDs) refer to a class of carbon-based nanomaterials, including graphene quantum dots (GQDs), carbon nanodots (CNDs), and polymer dots (PDs), inter alia, which have amorphous to nanocrystalline cores [1]. Before the discovery of CDs in the early 2000s [2,3], conventional dyes and semiconductor quantum dots were commonly used [4,5]. However, their applications are restricted due to the utilization of highly hazardous heavy metal ions [6], which has led to a shift in interest towards CDs and a thorough analysis of them. They have recently received significant attention because of their unique properties such as low toxicity, biocompatibility, chemical inertness, tunable fluorescence, flexible surface modification, and a wide range of applications, for example, chemical sensing [7], bioimaging, biosensing [8], nanomedicine [9], photocatalysis [10], drug delivery [11], fluorescent probes [12], and optoelectronic devices [13].

Currently, there are several methods available for synthesizing CDs. Generally, there are two strategies for synthesizing CDs: “top-down”, which is cutting from different carbon sources, such as graphite powder, carbon rods, carbon fibers, etc., and “bottom-up” synthesis from organic synthons or polymers and modification of surface functionality or passivation [1]. One step in the synthesis of CDs is the formation of fluorophores by the aromatization of the carbon cycles. These aromatization methods include pyrolysis synthesis [14], arc discharge [15], carbonization [16], solvothermal synthesis [17], hydrothermal [18,19,20,21,22,23,24,25], microwave [26], ultrasonication [27], electrochemical [28], and chemical oxidation [29]. CDs fabricated by those methods can be processed in large quantities, but there are limitations when it comes to producing spatially resolved carbon dots or dedicated 3D fabrication inside materials. Whereas embedding [30] or coating [31] can to some extent incorporate them into materials, these methods do not allow a spatial control of the CDs’ position.

IR femtosecond laser direct writing (FLDW) can produce spatially resolved CDs with a high degree of flexibility, aiming to overcome the limitations of traditional synthesis methods. This approach leverages the extremely high intensity at the focus of femtosecond lasers, leading to localized modifications through nonlinear absorption of near-infrared (NIR) photons. Such attributes render FLDW a promising technique for 3D functional implementation in chips and optical fibers, such as nanogratings [32,33] and nanocrystallizations [34], etc. Additionally, this method has already been used to create CDs in some materials such as PMMA, Polydimethylsiloxane, polycarbonate [35], Cyclo Olefin Polymer (ZEONEX^®^) [36], amino acid single crystals [37], and even cells [38].

In this study, we have demonstrated that IR femtosecond laser direct writing (FLDW) can produce spatially resolved CDs in the core of a perfluorinated fiber of CYTOP. CYTOP (cyclic transparent optical polymer [39]) is a promising graded-index fiber, exhibiting low attenuation at telecom wavelengths [40] due to the replacement of C-H bonds with fluorocarbon groups (C-F), making it highly suitable for optical fiber applications. In addition, CYTOP is less sensitive to environmental influences, with characteristics such as being insoluble in organic solutions and having low water absorption. The absence of residual optical absorption by CYTOP at the laser wavelength is compatible with the non-linear absorption processes that are readily induced using a femtosecond laser. The absence of CH bonds in the CYTOP implies that we have produced perfluorinated carbon dots and this is for the first time.

## 2. Experimental Details

### 2.1. Materials 

The fibers used in this research are amorphous perfluorinated plastic optical fibers (POFs), designed and produced by Chromis Technologies. The material type is CTL-S with -CF3 ends (refer to [39]). These fibers have a diameter of 90 µm and are composed of a pure CYTOP cladding around a 62.5 µm CYTOP graded-index core with dopants [41]. The over-cladding, made of XYLEX^®^ material (a blend of polycarbonate and an amorphous polyester), is removed by dissolving it in CH_2_Cl_2_ for a few minutes. The effective refractive index of these fibers is around 1.34 (at λ = 589 nm), with a higher index in the core. Further details are available in the product datasheet. CYTOP fibers exhibit high transparency in UVA, UVB, visible, and IR ranges up to 2000 nm. Specifically, the attenuation at 1030 nm is 30 dB/km and absorption starts from 190 nm (6.5 eV) [42,43]. The amorphous transition temperature is 108 °C and thermal decomposition starts from 400 °C. 

### 2.2. Carbon Dots Synthesis: Fs Laser Irradiation

For the CD fabrication, we used a Yb-doped fiber pulsed fs laser (Goji HP, Amplitude System, Pessac, France) operating at a wavelength of 1030 nm, with a pulse duration of 165 fs and an adjustable repetition rate (RR) ranging from 5 MHz to 10 MHz. The laser beam was focused into the core of the CYTOP fiber through an objective of 0.75 NA (40×). The average power of the laser on the samples was adjusted from 18.5 mW to 703 mW. The beam is linearly polarized along the fiber axis. Given that the CYTOP fiber has a graded refractive index (RI) around 1.3395 in near-IR, the fiber was immersed in a 5.6% Glycerol aqueous solution ($n_{D}^{20}$ = 1.3399) [44] as a phase-matching solution to minimize the influences of the fiber shape and interface on the focusing. The IR fs laser light diameter and divergence were adjusted using a beam expander (ZBE11, Thorlabs, Newton, NJ, USA) to firstly completely cover all the pupils of the objective and secondly to adjust the IR light focal plane to coincide with the observation of visible light. A shutter with an opening time of 100 ms was applied to control the irradiation exposure time. 

### 2.3. Raman Spectroscopy

To identify the nature of the species produced by the laser irradiation, Raman spectra of the CYTOP fiber in pristine and irradiated luminescent regions were measured. Measurements were conducted under a 780 nm continuous wave laser using a Raman Microscope system (DXR, Thermo Scientific, Waltham, MA, USA). The depolarized laser beam was focused on the sample with a microscope objective of magnification ×50 and NA = 0.5 or 0.75. The aperture of the pinhole was fixed to 50 µm, resulting in a spatial resolution of 1 µm in the focal plane and 2 µm in depth. Spectra were recorded from 3421 cm^−1^ to 63 cm^−1^ with an estimated spectral resolution from 4.7 cm^−1^ to 8.7 cm^−1^ for probe laser powers of 5 mW and 10 mW. Exposure time was set to be from 5 s to 15 s. The averaging was set to be typically around 25 times. The removal of the fluorescence background from 3421 cm^−1^ to 63 cm^−1^ was achieved by adjusting the baseline with a 6th order polynomial. The spectra were then decomposed through the principal component analysis (PCA) method.

### 2.4. Transmission Electron Microscopy (TEM)

The irradiated fibers were embedded in Epoxy resin (Kit LV premix Medium, Agar Scientific AGR1165, Stansted, UK) and polymerized for 24 h at 60 °C (a temperature below the operating maximum for CYTOP). Ultrathin sections were obtained with an ultramicrotome (UC6, Leica Microsystems, South San Francisco, CA, USA) and a 35° diamond knife (Diatome) through the fiber cross-section. Sections were collected on 200 mesh nickel formvar carbon-coated grids and were imaged with a JEOL1400 operating at 120 kV equipped with a RIO9 camera (Ametek, Berwyn, PA, USA).

### 2.5. PL Properties of CDs and Other Luminophores 

**A** 515 nm probing light is the Second Harmonic Generation (SHG) of the above irradiation IR laser, produced through a BBO crystal (β-BaB_2_O_4_) with a phase-matched angle. The probing light has the same parameter as the irradiation light, with a 165 fs duration and RR of 10 MHz. The probing light is expanded into a collimated beam when illuminating the sample. 

Lifetime and the corresponding intensity decay were recorded as an image by a space- and time-resolved single photon counter (Photonscore GmbH, Magdeburg, Germany). Samples were probed by the 515 nm SHG of the irradiation fs laser expanded with the parameter of RR = 10 MHz. Photons are saved with their attributes (time and place) allowing multiple analyses of the data [45]. In order to measure the environmental sensitivity of the luminescence, an O_2_ environment was created by placing the irradiated fibers in a glass bottle and flushing them with O_2_ for 20 min, to ensure a complete gas exchange. A no O_2_ environment was created by the same method but flushing with inert Ar gas.

Excitation spectra and emission spectra by UV excitation were measured by a spectrofluorometer (PTI QuantaMaster 8000). **A** 75 W Xe arc lamp from Quanta Master System (Horiba JY, Kyoto, Japan) was used as the excitation source, with a wavelength selected by a single monochromator. 

**The Excitation/emission matrix (EEM) in the visible range** were performed with a confocal microscope (Sp8-X, Leica, Wetzlar, Germany) equipped with a white light pulsed supercontinuum source based on Photonic Crystal Fiber, with a pulse duration of 120 fs and repetition rate of 80 MHz (NKT photonics, Brondby, Denmark). In this paper, the collection of light remained within a 12.5 ns window. The objective magnification was ×20 (oil immersion with 0.7 NA). The optical axial spatial resolution with the pinhole (53.1 μm) of the confocal microscope system using 10× objective was 565 nm at λ = 470 nm and 675 nm at λ = 670 nm. The excitation wavelength was varied from 470 nm to 670 nm with an 8 nm step. The emission spectra were measured with a Hybrid photodetector (Leica HyD) from 494 nm to 694 nm with an 8 nm spectral resolution. The Hybrid photodetector, combining PMTs and avalanche photodiodes (APDs), was used because it offers improved sensitivity (47% instead of 25%) and a lower level of dark noise compared to classic photomultiplier tubes (PMTs). The emission/excitation spectra matrix (EEM) was recorded for each pixel of the microscope image. However, small regions of interest (ROIs) were defined to perform local averaging to show the difference from one position to another. Specifically, the EEM visualized the energy level distribution of the luminophores. When associated with a principal component analysis, it gives the number of species, their associated spectra, and allows their localization to be identified [36].

## 3. Results 

### 3.1. Fs Laser-Induced Modifications

A series of irradiation experiments were conducted to explore the synthesis conditions of carbon dots. The fs pulsed laser was focused on a fixed position within the fiber core. During exposure to the laser light, the material experienced various changes. We established thresholds and processing windows for these different permanent modifications relative to laser power (*y*-axis) and exposure time (*x*-axis), as illustrated in Figure 1a. In these experiments, the repetition rate of the laser was set to 10 MHz, laser average powers were incrementally increased by 3.7 mW steps from 37 mW to 74 mW. The exposure times were set at 100 ms intervals starting from 100 ms up to 2.5 s, followed by longer intervals of 0.5 s until 5 s. This variation allowed for an escalating cumulative dose. After irradiation with these laser conditions, we identified three types of modifications, defined by their distinct morphologies, as shown in Figure 1b. 

**Type 1**. This is the first stage of visible modification when exposed to fs laser light. It is observable at a low power level, appearing as a black dot with the size of the beam waist diameter at 1/e^2^ (~1 µm), as shown in Figure 1b. During irradiation, this dot may exhibit slight movement within a 2 µm radius. 

**Type 2**. This stage is characterized by the formation of an ellipsoidal modified region, including a cavity (or void) in the center and a few rings of refractive index change. This region gradually enlarges over the exposure time, pushing material outward and resulting in the formation of concentric rings of refractive index change around it, as shown in Figure 1b. Occasionally, this process also produces a multifold spiral “galaxy” structure. This dynamic can be observed in a video in the Appendix A.

**Type 3.** This kind of modification occurs when the enlargement process of Type 2 continues to a point where the material cannot withstand the internal pressure and overcomes the rupture limit.

This observation suggests that the fs laser induced some gas to form a Type 2 cavity. From a top view of a Type 3 modification, as shown in Figure 1b, the central black region is a tunnel rather than a spherical cavity. Inside this tunnel, structures resembling a lamprey’s mouth are observed, which may be indicative of rapid crystallization.

Figure 1a illustrates the boundaries and processing windows of these three modification types. Below 40 mW (pulse energy 4 nJ at repetition rate 10 MHz), no visible transitions are observed, regardless of the exposure duration. In this parameter condition, especially for RR of 10 MHz, the transition of Type 1 to Type 2 occurs rapidly. Consequently, the window to generate Type 1 is narrow, appearing merely as a dividing line between the non-modified region and Type 2 modification. It corresponds to a dose of 5 mJ, which is shown in Figure 1a by a thick green line. Type domain can be adjusted approximately by the function $\mathrm{Power}\left(\mathrm{mW}\right)=39\;\mathrm{mW}+\frac{4\mathrm{mJ}}{\mathrm{exposure}\;\mathrm{time}\;\left(s\right)}$. The size of the modified regions of Type 2 seems to linearly increase over exposure time in Figure 1c. Type 3 modification boundaries can be adjusted by $\mathrm{Power}\left(\mathrm{mW}\right)=44\;\mathrm{mW}+\frac{4\mathrm{mJ}}{\mathrm{exposure}\;\mathrm{time}\;\left(s\right)}$. The equation can also be interpreted as being composed of an incubation dose (4 mJ) and then a dose debit (39 or 44 mW) depending on the modification type finally achieved. Thresholds in Figure 1a are a reference (i.e., magnitude and trend) and may be shifted upwards overall due to environmental perturbations (beam quality and fiber surface).

### 3.2. Photoluminescence (PL) Properties

**3D distribution and excitation/emission matrix in the visible range.** Photoluminescence (PL) was detected in Type 2 and Type 3 modifications, indicating a chemical change in the molecular structure of the fiber. In the case of Type 2 modification, an increase in PL intensity is observed with the accumulation of more energy over time. However, prolonged exposure time or higher power caused the Type 2 modified region to break, forming a Type 3 tunnel. In many instances, Type 3 displayed lower PL intensity compared to some large-size Type 2 modifications. This decrease may be due to the expulsion of luminescent matter during breakage. Note that in some cases, Type 3 formed significantly faster under higher power, leaving insufficient time for the generation of luminescence species. This observation implies that optimal PL in Type 2 modifications can be achieved by fine-tuning the laser parameters and timing just before the transition to Type 3. Therefore, a “combo” method was developed. This method involved initially applying a power during a time minimum to trigger Type 2 modifications. Then, with a sufficient time delay, e.g., 30 s, power was reduced, and exposure time was distributed over multiple periods to gradually deliver energy. 

Figure 2a,b illustrate the PL distribution of two Type 2 ROIs created under different laser parameters. The recorded confocal microscopy image and its corresponding transmission image under natural light are shown in Figure 2b and Figure 2a, respectively. ROI1 is a standard Type 2 modification (irradiated at 10 MHz, 51.8 mW for 2 s), while ROI2 is a Type 2 modification using the ‘combo’ method (10 MHz, 51.8 mW for 2 s, followed by two instances of 18 mW for 100 ms), denoted Type 2+. This method enlarges the void and significantly increases PL intensity, i.e., 10 times greater than typical Type 3 modifications. Figure 2c displays a screenshot of the top and side views of a 3D reconstruction of these two luminescent volumes, measured along the *z*-axis (10 nm step). From Figure 2b,c, we observe that in ROI1, the PL is distributed throughout the cavity with a slight layer texture, while ROI2 exhibits higher intensity around the periphery compared to the center. Note that in the side view of the 3D images, the PL distribution is asymmetric along the vertical direction; this is an artefact of confocal microscopy, as the signal was disrupted during the measurement of planes behind the void. Therefore, the PL distribution in Type 2 and Type 2+ modifications are ‘solid’ and hollow ellipsoid, respectively. Since the images were measured simultaneously, it is evident that the ‘combo’ method created greater PL in ROI2. A similar PL created by fs laser irradiation was observed in polyimide [46], attributed to deposition on the inner surface after evaporation. However, in our study, the luminescent regions appear to be on the outer surface of the void by comparing them with the images in Figure 2a,b, as further corroborated by TEM results. 

In addition, the two ROIs were spectroscopically studied in the visible range, as illustrated by two excitation/emission matrix (EEM) graphs in Figure 2d,e. These graphs reveal the presence of a primary species excited in the range of 470–490 nm (red center) in both EEMs, thus giving rise to broadband luminescence from green to red. Upon the injection of additional laser energy, ROI2 exhibits a newly generated luminescent center excited at 505 nm (see the cross in Figure 2e), with emission centered at 560 nm. This phenomenon is similar to fs laser-induced photoluminescence in Zeonex polymer [36]. Practically, it implies that color alteration can be achieved to a certain extent by modulating the ratio of luminescence excited at 500 nm through laser parameters.

**Luminescent species excited in the UV range.** To provide more comprehensive spectral information and to explore potential applications of the luminescent material across different light ranges, we characterized the performance of ROI2 in the UV region. Results are shown in Figure 3 and reveal one species excited at 400 nm and emitting light peaking at 474 nm. 

**Spatially resolved lifetime of irradiated Type 2.** Lifetime measurements serve as a tool to assess variation in the fluorescence yield of the same sample under different conditions of preparation and environment. In Figure 4a, the spatial distribution of the fluorescence lifetime (FLIM) highlights a distinct contrast between the center and the edge of Type 2 fluorescent spots (ROI2 in Figure 2). Specifically, the center displays an average lifetime of 2.5 ns, which increases to 3.5 ns towards the edge, as depicted in Figure 4b. We noticed a reproducible and stable fluorescence decay across all samples, with the decays exhibiting multiexponential characteristics as demonstrated in Figure 4c. This behavior can be attributed to two main factors. Firstly, the fluorescence may originate from a mixture of different species, as in the case of fs laser excitation of electrons, which subsequently recombine to produce carbon dots, which may be of multiple sizes over a range; their lifetimes span from 6 ns down to 2 ns. Secondly, the fluorescent molecules are surrounded by non-fluorescent polyaromatic molecules located at fixed random distances from them. Therefore, this variation could be due to the different nature of the fluorophores formed at different locations in the spot, or to the difference in the concentration of quencher in the center and the edge.

**Spatially resolved lifetime during irradiation by fs laser in scanning mode.** To acquire additional insights into the process of luminophore generation, we recorded the lifetime distribution during the fs laser irradiation process. Differing from irradiation in static mode, the sample was moved from right to left at a speed of 0.1 mm/s (refer to Figure 5), while simultaneously being probed by a 515 nm pulsed light, covering the entire sample. The lifetime image, shown in Figure 5a, displays a trace with a ‘hot spot’, corresponding to the laser focal volume. The entire image was collected over 12 s, as we can see a trace of 1.2 mm in length. PCA analysis reveals that three decays could describe 95% of the luminescence, as depicted in Figure 5b. With the spatial distribution of these three components illustrated in Figure 5c, we deduce that: (1) The red profile corresponds to luminescence induced by the IR fs laser through a multiphoton excitation process, starting 1.08 ns before the other two contributions, with an average lifetime of 7.8 ns. (2) The green curve represents the luminescence of the trace excited by the 515 nm laser light, modified by fs laser shortly before, with an average lifetime of 21 ns. (3) The blue curve indicates a contribution constantly located around the IR focal point over 10 µm, excited by the 515 nm laser light, with an average lifetime of 4.8 ns. It is labelled as ‘hot’ because it appears only around the focus of the laser beam. Figure 5d shows the intensity of these three components during the irradiation process, measured in photon/s (shown as kHz). The IR fs laser was activated at 1 s, represented by a sudden increase in IR-induced luminescence (red curve) and the blue curve. Note that in Figure 5d, the blue curve experienced a sudden increase at around 7–8 s, lasting around 1 s, possibly due to a temporary accumulation triggered by defects.

**Stabilities in O_2_ environment.** Type 2 modifications were embedded within the fiber, isolated from the environment, rendering them insensitive to environmental influences such as pH and oxygen, while Type 3 modifications had an exit connecting to the exterior. It is widely acknowledged that exposure to an oxygen atmosphere can often quench luminescence, which is a critical consideration in practical applications like OLEDs, where creating an oxygen-depleted environment might be necessary. We thus investigated the sensitivity of this luminescence to O_2_. For this purpose, we measured the lifetimes of a Type 3 modification which is exposed to the environment, in both an O_2_ and Ar atmosphere. The decay profiles are displayed in Appendix B Figure A1. The result reveals a negligible difference under air, Ar, and O_2_ environments, suggesting that the observed luminescence is not significantly quenched by oxygen. This inference is drawn from the decay profile of air, which does not position itself between the decay curves of the Ar and O_2_ environments. These laser-induced fluorescent carbon dots are insensitive to the presence of O_2_, indicating their potential utility in applications where exposure to oxygen is unavoidable.

### 3.3. The Identification of Luminescent Region Containing Carbon Dots

The utilization of femtosecond (fs) laser irradiation in the transformation of molecules within a polymer composed solely of carbon and fluorine, resulting in luminescent properties, strongly suggests the formation of carbon–fluorine compounds with aromatic structures. This aligns with numerous studies that have explored the effects of fs laser irradiation on other organic polymers [38,47]. To corroborate our position and further understand the molecular nature of the products formed by laser treatment, we conducted Raman spectroscopy and Transmission Electron Microscopy (TEM) analyses.

**Raman spectroscopy.** Figure 6 displays the Raman spectra of the CYTOP fiber core and the irradiated region corresponding to Type 2 modification, with an insert image showing the measured positions. In the low-wavenumber region of the CYTOP core spectrum (70–800 cm^−1^, orange profile), the strongest peak at 690 cm^−1^ is attributed to aliphatic F-C-F vibrations [48], accompanied by two peaks within 250–350 cm^−1^. Other peaks in the low-wavenumber region, plus two peaks at 1450 and 1675 cm^−1^, are ascribed to dopants that contribute to the index gradient, and details are described in Appendix C The Raman spectrum of the Type 2 modified region (blue profile) is characterized by a significant presence with two maxima at 1315 cm^−1^ and 1600 cm^−1^ corresponding to the G-peak and D-peak of a graphene/graphite carbon-based structure [49,50]. The G-peak, typically observed at 1580 cm^−1^, is associated with the stretching of sp2-hybridized carbon bonds in a plane within graphitic structures. The D-peak, occurring at 1350 cm^−1^, arises from the breathing mode of sp3-hybridized carbon atoms in disordered or amorphous carbon structures. However, a slight shift and broadening of these two peaks were noted in our samples, which we hypothesize may be due to the chemical environment such as the presence of fluorine atoms. Similar spectral characteristics, including a stronger D-peak relative to the G-peak, a vanishing of the 2D peak, and a broadening of both the D-peak and G-peak, have been reported in oxidized [49,51] fluorinated graphene during the fluorination process [52].

**Transmission Electron Microscopy.** To elucidate the structural changes induced by femtosecond laser irradiation and understand the formation of luminescent properties within the CYTOP polymer, Transmission Electron Microscopy (TEM) analysis was performed. This method is critical for providing direct visual evidence of microscopic changes and for validating the nature of modifications at the nanoscale. CYTOP samples were prepared by cutting into thin slices of approximately 70 nm. The images shown in Figure 7 offer detailed insights into the modifications. Figure 7a displays an overview of a Type 2 modified region, where brighter ellipsoids appear, indicating the formation of voids. Figure 7b–d provide zoomed-in views of different regions marked in Figure 7a, while Figure 7e is a zoom of a region indicated in Figure 7d. These observations reveal the presence of black dots and voids surrounding the laser-induced void in the Type 2 modification. Those small voids around the central void are indicative of gas production during the irradiation. Notably, the black dots appear smaller, fewer, and shallower the further they are from the center until around 1–2 µm in the periphery. This observation suggests that the luminescence is not a deposition on the inner surface of the void but rather extends around the periphery, within a layer approximately 1–2 µm thick. Figure 7e displays the details of some black dots, with the size ranging from a few nanometers to tens of nanometers. 

## 4. Discussion

**Kinetic processes of laser–material interaction and carbon dot generation processes.** In the discussion, we delve into the kinetic processes involved in the interaction between the laser and material, focusing on the generation of carbon dots within CYTOP, a material with low absorption at 1030 nm. Our observations lead us to propose a scenario: when a laser irradiates the sample, at the focal point there is an extremely high intensity, and the material partially absorbs the pulsed laser energy through multi-photon ionization [53], subsequently leading to the ionization of CYTOP [53]. After several hundreds of milliseconds, a black dot appears at the focal point through natural light transmission observation. This modification that we have named Type 1 likely results from the transformation of the solid state to a more plastic phase, as we observed a movement of the black dot. With further energy injection, gas generation occurs within milliseconds, transforming the black dot into a void. This void finally increases in volume over irradiation time, and the appearance of rings attributed to a sudden volume increase is noted (likely a shock wave) generating a Type 2 modification. Concurrently, carbon dots are formed and distributed, maybe diffused outward around the periphery of the void. Over an extended period, more gas is produced, and its expansion exerts a high pressure, enlarging the voids and affecting the surrounding density, which may lead to material fracturing. Occasionally, a multifold spiral “galaxy” structure is observed through natural light transmission, revealing some twisting of the matter. The origin remains under investigation. Some crystals also appear on the internal surface of the cavity, indicating the occurrence of a low viscous phase in the void. Ultimately, this process culminates in a breakthrough to the fiber surface, forming a Type 3 modification, with the emergence of a bubble from the newly created tunnel. A comprehensive video detailing this process is available in the Appendix A.

Specifically, in Type 2 modification, the void size increases over exposure time before the transition to Type 3. The increase is almost linear, as shown in Figure 1c. The formation of a void in transparent materials with femtosecond laser irradiation is a complex multi-physical phenomenon caused by the generation of locally dense plasmas whose recombination brings matter into a highly stressed state. The geometry of the heat-affected zone and the state of the matter in this region (softening, melting, or vaporing) depend on the irradiation parameters where the focusing geometry plays an important role. The most intuitive image of a void formation as Type 2 modification (shown in Figure 1b) could be the laser-induced excited electron plasma production that triggers the vaporization of the material in the center, as we have seen the escape of the gas in the latter stages of the process, and melting or softening around the heat-affected zone, and waveform-like refractive index change. The process is also described in this way under the interaction of a femtosecond laser with polyimide [46]. In addition, some other void formation dynamics have been studied in inorganic materials such as silica glass materials, e.g., [54,55,56]. Depending on the materials and irradiation conditions, the formation of cavities is attributed to different scenarios such as spontaneous bubble nucleation in the molten phase [57], high pressures induced by focused laser beams [58], the presence of interstitial oxygen, tensile stresses in the liquid melt, rapid quenching, and so on [55]. In these processes, the temperature distribution appears determining. We thus examine this point below.

**Thermal model of laser–matter interaction.** After the laser energy deposition at the focus, a heat flow to the surroundings occurs in a few µs. This thermal process can help in understanding the modification induced by fs laser pulses. Here, we computed the temperature fluctuation and evolution over time (until 10 µs) at the center and periphery of the beam focus in CYTOP material caused by multi-pulse energy absorption, using spherical source modelling [59]. Results are shown in Figure 8a,b. Significant heat accumulation in CYTOP compared to silica glass is expected due to its relatively smaller thermal conductivity. The energy deposition process is considered to be instantaneous. Since the difference between the maximum temperature (caused by a pulse deposition) and the minimum temperature (after thermal diffusion within the pulse period) is very small for this RR, we can thus consider a time-averaged temperature everywhere as a reasonable approximation. Figure 8c gives the average temperature distribution for different laser powers when the thermal system gets to the steady state. 

During Type 1 and Type 2 modifications, we observed that the material exhibits viscoelastic and plastic behaviors during irradiation, evidenced by the enlargement of the cavity and displacement of the Type 1 dot. The temperature distribution curves at steady state induced by varying powers are plotted in Figure 8c. The amplitudes of these distributions were based on an absorption rate at A = 0.005, chosen for its alignment with the experimental power threshold and the material’s key thermal properties, such as glass transition temperature (T_g_ = 381.15 K) and decomposition temperature (T_decomposition_ = 673.15 K [42]), shown in black dash lines. With powers of 37 mW, 55.5 mW, and 74 mW, the temperatures induced by the laser pulses result in no material modifications, melting, and chemical reactions, respectively. However, this temperature induced by laser pulses can explain the modification threshold but not the reasons behind the cavity size change over the exposure time. In Type 2 modification, the void size and its exposure time dependence are not directly related to the temperature-affected zone, because firstly, a steady state is reached in less than 3 ms (calculation details are in Appendix D and [59]), and the exposure time arising from 500 ms to 1 s should not change the laser-induced temperature. Secondly, the void size is larger than the heat-affected width shown in Figure 8c. Therefore, this may indicate that new molecules induced by the laser can continuously absorb the laser energy to produce a new round of temperature increase, and if it contains gas, the volume is enlarged. Figure 8d,e show at a larger time scale a temperature evolution until 100 s based on this process; Figure 8a is involved in this profile as an invisible step from zero to the lowest value at the beginning (0–10 µs). Then, we have assumed a simple laser-induced chemical reaction from A to B_i_ with a constant rate k_0i_ and activation energy Ea_i_ that allows placing the half-time of the reaction (advancement degree = 1/2) in the second time scale, as shown in Figure 8d,e. Figure 8d,e display the temperature evolution induced by these chemical reactions according to reaction rate k_0i_ and activation energy Ea_i_. Results demonstrate that both Ea and k_0_ influence the incubation time, while the temperature slope is influenced by Ea.

**Commonality on spectral properties**. We found a specific distribution of luminescent species across several laser-modified organic materials, including glycine single crystal [37], Zeonex [36], and even caramelized sucrose. This finding suggests a commonality in chemical structures of the luminescent species generated by a femtosecond laser in these organic materials, possibly related to a similar size of carbon dots. The presence of fluorescent species at 475 nm excitation indicates that the size of the carbon dots produced falls within a favorable range. This is supported by reports showing that the HOMO-LUMO gap is strongly affected by the size of GQDs [61,62,63]. TD-DFT calculations revealed that the absorption wavelengths of the pristine GQDs vary with size, ranging from 168.06 nm to 507.86 nm, correlating with sizes from 0.27 nm to 1.43 nm, or from C6 to C76 [62], consistent with experimental observation [64,65]. Based on that, the carbon dots with absorption at 400 nm, 475 nm, and 520 nm produced with higher energy deposition are estimated to have sizes around 1.1 nm, 1.4 nm, and 1.5 nm, with approximately 5 phenyl rings. In addition, Ma et al. [66] separated caramel into low- and high-molecular-weight portions by dialysis, discovering blue fluorescence under UV light in the former, which indicates that the fluorescent species are small molecules. However, further investigation is needed to understand why laser-induced species in organic materials tend to be of this specific size range.

## 5. Conclusions

CYTOP fiber, composed of only carbon, fluorine, and oxygen atoms, exhibits low attenuation in the visible–IR range compared to other polymers. We have successfully synthesized luminescent perfluorinated CDs within the core of this optical fiber for the first time, through a simple, efficient light-based method, known as FLDW. We have identified an optimal set of laser parameters and developed a ‘combo irradiation’ that enables the production of luminescent species with good efficiency. These laser-induced CDs have at least three excitation centers from the UV to visible range, exhibiting efficient emission from the green to the red spectral range. Additionally, the light dose influences not only the PL intensity but also the cavity size, which can be utilized to control optical polarization. Our thermal and chemical modelling indicates that the cavity size, itself, is influenced not only by laser energy deposition but also by the induced species absorbing the laser light more efficiently through subsequent chemical reactions, thereby increasing the temperature over a larger radius around the initial voids. Moreover, the functionalization of CYTOP optical fibers with CDs holds promise for applications such as luminescence-based fiber sensing, stimulators, and UV down-converters. Its insensitivity to oxygen may contribute to achieving durable devices. Furthermore, our study highlights the potential of FLDW as an eco-friendly and efficient processing technique for selectively synthesizing CDs with novel properties and applications.

## Figures and Tables

**Figure 1 nanomaterials-14-00941-f001:**
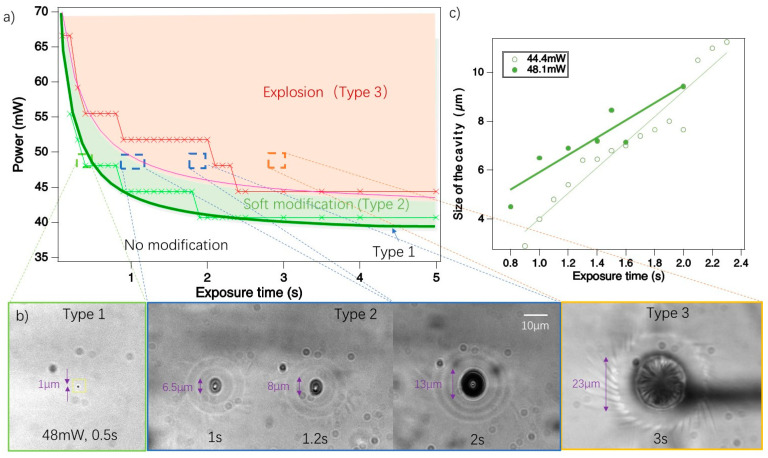
(**a**) Threshold profiles and process windows of modification types; green for Type 1 and 2 and red for Type 3. Dark green and pink curves indicate the fixed dose of around 4 mJ with offset 39 and 44 mW, respectively. Small crosses mark experimental data. (**b**) Microscope images of modification types in transmission mode using natural light. Modification parameters correspond to dashed frames in (**a**). (**c**) Cavity size and linear fitting of Type 2 modifications versus exposure time for laser powers of 44.4 mW (green) and 48.3 mW (dark green).

**Figure 2 nanomaterials-14-00941-f002:**
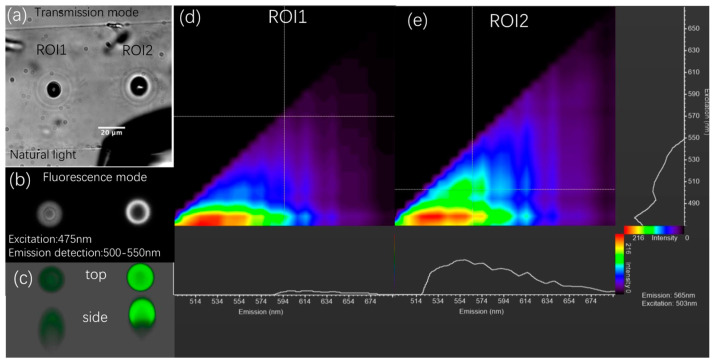
(**a**) Transmission image of two laser-irradiated regions. Parameters of ROI1: 1030 nm, 165 fs, 10 MHz, 51.8 mW, polarization as shown in (**a**), exposure time of 4 s; ROI2: same as ROI1 except for two instances of extra exposure of 100 ms under 18.5 mW. (**b**) Fluorescence image of ROI1 and 2. (**c**) Screenshots of 3D model of ROIs on top view and side view. (**d**,**e**) The excitation/emission matrix of ROIs, with a 470–670 nm range for the excitation and a 494–694 nm range for the emission. The color stands for the intensity and interpolation processed by the software, and raw data can be found in complementary information.

**Figure 3 nanomaterials-14-00941-f003:**
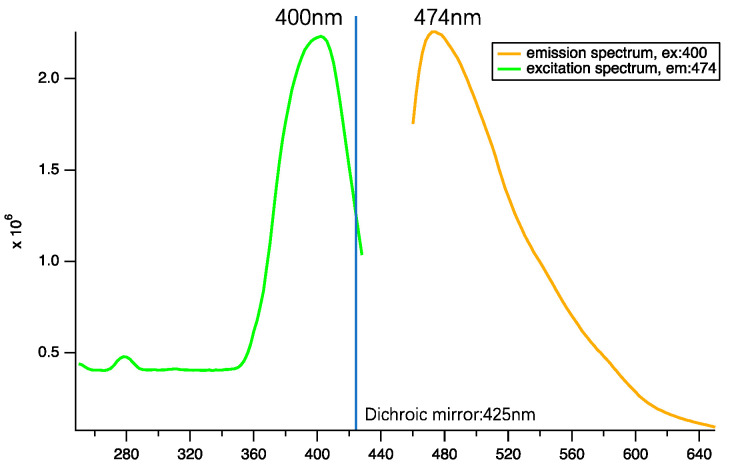
Excitation spectrum in UV range when detected at 474 nm (green) and emission spectrum when excited by 400 nm continuous wave laser.

**Figure 4 nanomaterials-14-00941-f004:**
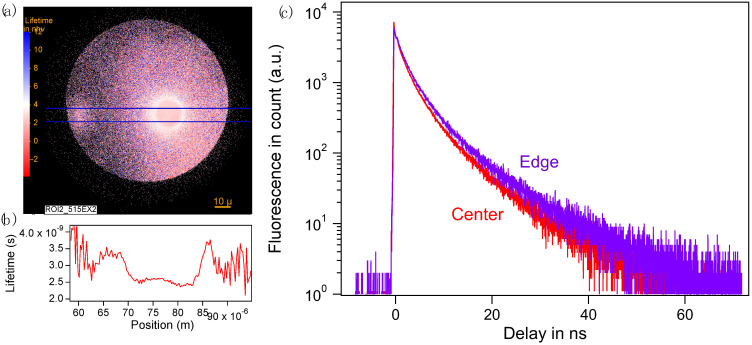
(**a**) Lifetime distribution of ROI2 and (**b**) lifetime distribution of ROI2 along the blue horizontal line marked in (**a**). (**c**) The fluorescence decay of the center (red) and edge (purple) of ROI2.

**Figure 5 nanomaterials-14-00941-f005:**
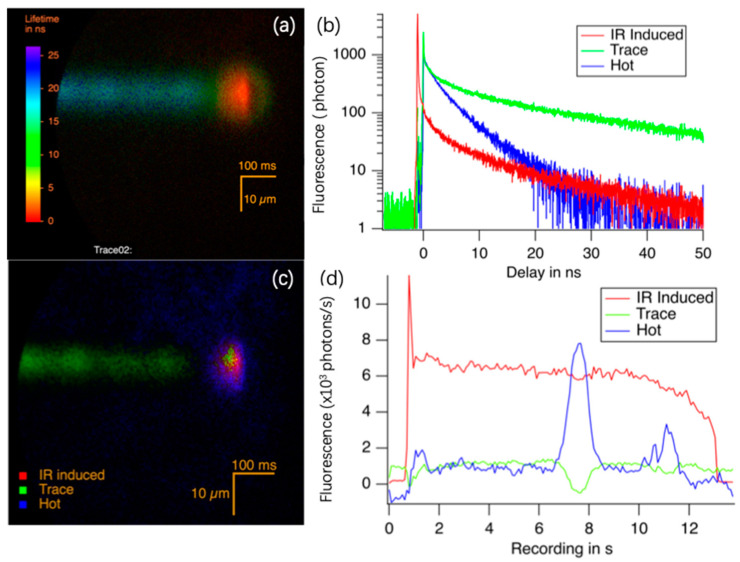
FLIM at the place of irradiation. (**a**) Lifetime distribution during fs laser irradiation. Laser parameters: 5 MHz, 40.7 mW, and scanning speed 100 µm/s. The laser beam focus is located on the right side (shorter lifetime with red color); (**b**) the 3 first components of (**a**); (**c**) spatial distribution of these 3 components; (**d**) intensity of these 3 components during the irradiation process.

**Figure 6 nanomaterials-14-00941-f006:**
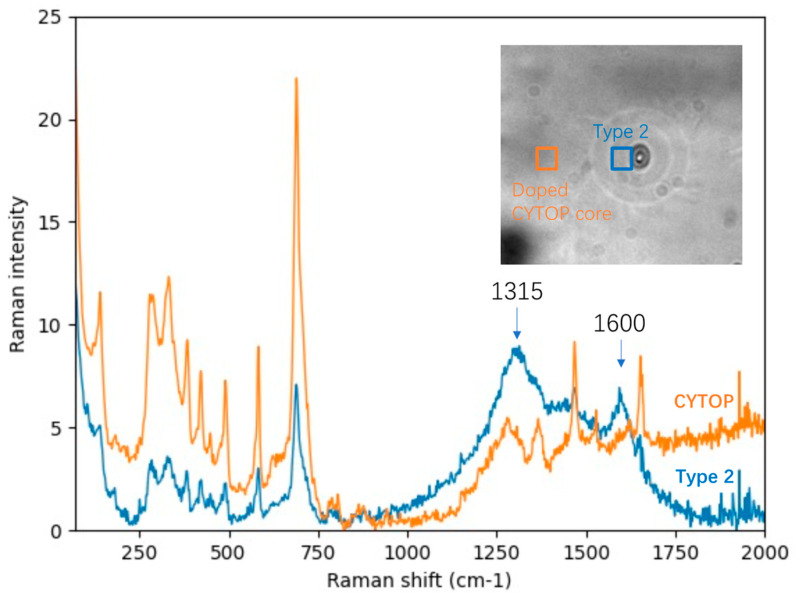
Raman spectra of pristine CYTOP fiber core (orange) and the irradiated region of Type 2 modification (blue). Insert: the measured areas. Laser power for Raman spectroscopy was set to be 10 mW, with 20 s exposure time and 16–25 instances of acquisition, using objective 50× with NA = 0.5.

**Figure 7 nanomaterials-14-00941-f007:**
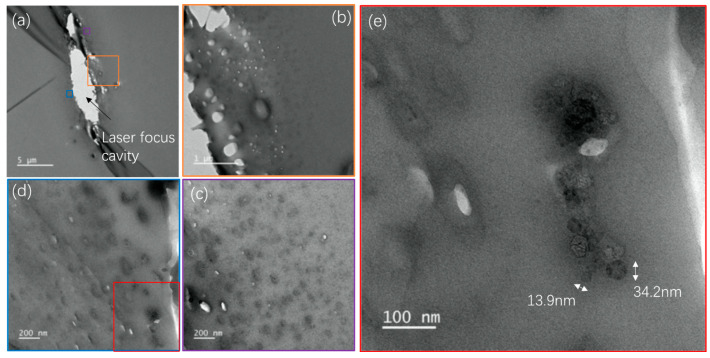
TEM images of Type 2 area irradiated by fs laser of 74 mW and 10 MHz. The fiber axis is perpendicular to the plan of the photo. (**a**) A typical cavity in the Type 2 modification region. (**b**) Enlarged image of the area indicated by the orange square in (**a**). (**c**) Enlarged image of the area indicated by the purple square in (**a**). (**d**) Enlarged image of the area indicated by the blue square in (**a**). (**e**) Enlarged image of the area indicated by the red square in (**d**).

**Figure 8 nanomaterials-14-00941-f008:**
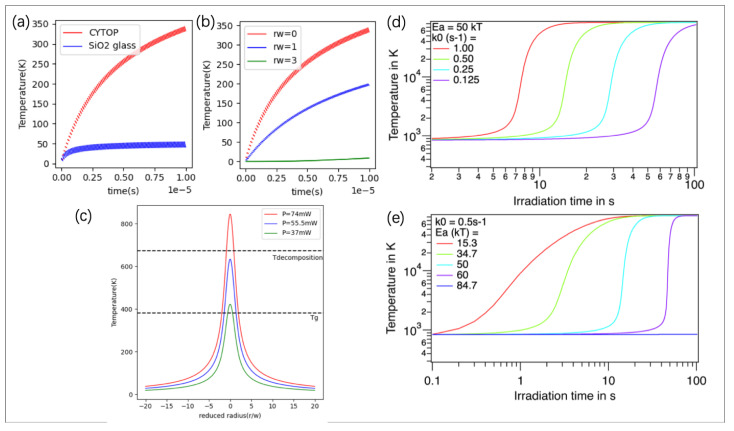
(**a**) Temperature evolution over time (the origin is room temperature) by 100 laser pulses in a frequency of 10 MHz in CYTOP and SiO_2_ glass using spherical source modeling [59]. The parameters of materials are: CYTOP: $\kappa =0.12\;W/\left(m\cdot K\right),\;\rho =2030\;\mathrm{kg}/m^{3},\;C_{p}=861\;J/\left(\mathrm{kg}\cdot K\right)$ [42]. SiO_2_ glass: $\kappa =1.38\;W/\left(m\cdot K\right),\;\rho =2203\;\mathrm{kg}/m^{3},\;\;C_{p}=703\;J/\left(\mathrm{kg}\cdot K\right)$ [60]. Other parameters for both materials: absorption rate A = 0.005, laser power 55.5 mW, and beam radius = 0.5 µm. (**b**) Temperature evolution over time at the beam center (reduced radius *r_w_* = *r*/*w* = 0) and around the center, at *r_w_* = 1 and 3. (**c**) Temperature distribution at steady state with laser power of 37 mW, 55 mW, and 74 mW. (**d**,**e**). Temperature evolution after steady state due to absorption by new species produced later (second is the time scale), after the initial thermal steady state is reached (the scale is the ms), with (**d**) same activation energy E_ai_ and different reaction efficiency k_0i_ and (**e**) different activation energy and same reaction efficiency.

## Data Availability

Data are contained within the article and Appendix A.

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
