# Peer review of "Carbon Dot Synthesis in CYTOP Optical Fiber Using IR Femtosecond Laser Direct Writing and Its Luminescence Properties"

_nanomaterials, 2024, doi:10.3390/nano14110941_

Round 1

Reviewer 1 Report

Comments and Suggestions for Authors

The authors present the results of the successful synthesis of luminescent perfluorinated CDs inside the core of this optical fiber using direct femtosecond laser writing. These results are original and new, but the text of the article requires additional explanations to continue a full analysis of the conclusions in the “Discussion” section.

1) In the description of Raman spectroscopy methods, high probe power (5W-10W) is indicated. In Raman measurements, such power can lead to modification of the sample. I think there may be a misprint here. If it is not, then comments are required on why such a high pump power is used. With focusing NA=0.5, 5W power is enough to ablate diamonds, metals, not to mention polymers and polymer fibers. (lines 104-105).

2) Was oil immersion used for the 0.7 NA objective in Excitation Emission Matrix measurements? If not, then the real numerical aperture of the lens is not 0.7 NA? Was this point taken into account when calculating/specifying the spatial resolution of the optical system?

3) Please, indicate the laser pulse repetition rate for laser modification modes in "Fs laser Induced Modifications" subsection. 

4) It is required to explain in more detail the values of the radiation dose and the expression for Power of the form “Power(mW)=39mJ+4mJ.s/exposure time(s)”. Where did 39 mJ and 44 mJ come from? In general, when considering the effect of laser radiation on a material, it is better to indicate the energy density (fluence) or pulse energy with an indication of the repetition rates. The text about fig. 1a and the caption to it require significant and more detailed explanations.

5) Next, from lines 210, the authors talk about structures of the second type (type 2). The parameters for their formation are also indicated: average power 51.8 mW with 2s exposure for Type2 and combo method for Type2+. When creating a type2+ structure, what is the time between primary irradiation (51.8 mW) and secondary irradiation (18 mW)? Why do type2 and type2+ structures form in these cases, if for the specified laser radiation parameters a type3 structure should be formed, following the graph in Fig. 1a? Some clarification required.

6) In Figure 2d, the axes and their marks are poorly visible. It is worth specifying the initial value of the “Emissions” axis report.

7) What material or structure (ROI1, ROI2) is studied in the section “Luminescent species excited in UV range”?

8) It is strange that in the “Spatially resolved lifetime of irradiated Type 2” section, Figure 4c appears first. You can relabel 4c as 4a and make appropriate changes to the figure. Has the kinetics of ROI1 and ROI2 been compared? Axes units fig. 4b can be safely changed to nanoseconds and micrometers.

9) Specify the parameters of 515 nm laser radiation for Spatially resolved lifetime measurements.

The results presented in the article are interesting. However, the article requires significant corrections for publication in Nanomaterials

Author Response

Please find the coverletter attached.

Reviewer 2 Report

Comments and Suggestions for Authors The manuscript under review suggests novel interesting results concerning the femtosecond laser-induced space-selective fabrication of carbon dots and space-selective intitiation of wide-range luminescence ability in the CYTOP optical fiber. The topic is well in the scope of the journal. The study combines advanced experimental analysis with the thorough numerical analysis of the obtained data and numerical modeling. The scientific level of the study is high as well as that of the data presentation.  I can surely recommend this paper for publishing after some questions are addressed and some drawbacks or vague poits are revised.   - What software was used to apply PCA method for decomposing the spectra and the decay curves? What software was used for laser-induced heating modeling?   - line 173: "This observation suggests that the fs laser induced some gas to form a Type 2 cavity."  Is Type 2 meant here indeed? In this case, you assume that Type 2 also comprises a cavity but if so then this is not evident from the Type 2 discription above and should be indicated and justified more clearly. Then it is also not clear why Type 3 is distinguished as a modification type in which the rupture limit is exceeded if a cavity appears already in the Type 2 modification. As it can be deduced from the further sections of the manuscript Type 3 exposes the enlarged cavity to the surface of the fiber and it is definitive for this type, isn't it? If so, it should be indicated more clearly in this section too.   - The polarization plane of the writing laser beam is mentioned to be oriented along the fiber axis. Is the orientation of the polarization plane significant for this experiment? Does it somehow influence on the modification? Did the authors check it or can you make any assumptions?   - The value whose plots are shown in Fig.5(d) is called intensity but is measured in photon/s and represented in kHz. First, using kHz here looks a bit confusing as Hz are usually referred to frequency-related or spectral values rather than to energy-related. Second, photon/s appears to be a unit for power rather than for intensity (i.e. power surface density). Was the luminescence signal collected from the same areas for all three curves? Can it be regarded and compared as intensity or is it more correct to consider it as power?   - Regarding the experiments on the stability of CD luminescence in O2 and Ar environments, how long were the luminescent areas exposed to those environments? Can the revealed stability expected to be a long-term one?   - One more question concerns potential applications of the suggested CDs fabrication method. The authors claim that it can be useful for functionalization of CYTOP optical fibers for fiber sensing and so on. However fs laser-induced CDs formation is accompanied with the emerging of um-sized cavities and significant local refractive index modification, which is expected to greatly affect the waveguiding properties of the optical fibers. This aspect should be somehow discussed.   Fig. 2: I recommend increasing scale subscriptions in Fig. 2(e). Now they are hardly readable even after zooming the image. Fig. 8: the fonts of the subscriptions to the Fig. 8(c) graph should be increased. Temperature units in Fig. 8(d,e) are missing and should be added.   Comments on the Quality of English Language Though the language of the manuscript is generally clear and correct, I recommend to check it once more because some minor grammar errors or misprints are still present. Some of them, which I could notice, are following: line 47: "CDs fabricated by those methods can be process in a large quantity" - it looks like "process" should be replaced with "processed" or some other correction is required. line 95: "opening" would be correct rather than "openning"  line 103: "were recorded" is expected instead of "were recorded" line 103: "were embedded" is expected instead of "were embedding" line 303: "can often quenches" is to be replaced with "can often quench", isn't it?

Author Response

Please find the attached coverletter.

Reviewer 3 Report

Comments and Suggestions for Authors

This paper focuses on creating carbon dots in CYTOP optical fibers, with data meticulously collected to structure the paper. However, the following points should be added and clarified.

There is insufficient explanation regarding why carbon dots are being produced on CYTOP fibers, i.e., the significance of this research is not adequately explained. If the application using luminescence is a potential application, it would be preferable to disperse and generate small carbon dots "only" instead of creating a large modification area at the core as done in this study.

Regarding the repetition rate of the laser in the experimental setup, while it is mentioned in the Experimental Detail section as “ranging from 5 MHz to 20 MHz”, it is not specified which repetition rate was used for each data point. Repetition rate is crucial for discussing modification as it greatly affects interactions, particularly heat accumulation. Moreover, it is essential for ensuring reproducibility in the paper for others.

In the discussion, separate discussions are necessary regarding the influence of initial modification and subsequent pulses. The absorption coefficient differs significantly before and after initial modification occurs. Additionally, reverse Bremsstrahlung occurs when plasma is generated.

Regarding the TEM images, when discussing the generation of carbon dots, higher resolution and magnification images should be provided, showing lattice fringes. Since various forms of carbon are likely present in this study, data such as Raman spectra may not exclusively originate from dot-shaped materials. To conclusively demonstrate the production of carbon dots, lattice fringes are necessary.

Minor comment:

Some abbreviations (PC, PDMS, etc.) need to be spelled out for clarity.

Comments on the Quality of English Language

well written

Author Response

Please find the attached coverletter.

Round 2

Reviewer 1 Report

Comments and Suggestions for Authors

Thanks to the authors for their responses to the first review, but some moments still need to be corrected.

1. Consider the equation for power presented in the answer and added to the text (lines 207-212): 𝑃𝑜𝑤𝑒𝑟(𝑚𝑊) = 39𝑚𝐽 + (4𝑚𝐽/𝑒𝑥𝑝𝑜𝑠𝑢𝑟𝑒 𝑡𝑖𝑚𝑒 (𝑠)). It does not converge in dimension. I assume that instead of 39 mJ there should be 39 mW and then 44 mW instead of 44 mJ.

2. Line 236, subsection “3D distribution and Excitation Emission Matrix in the visible range”. The authors consider the formation parameters of the Type 2 (ROI1) structure of 51.8 mW with an exposure time of 2 seconds. Following the boundaries indicated in Figure 1, a structure of type 3 should be formed in this regime. This moment is not clear.

3. The caption of Figure 2 shows the parameters of ROI1: 1030nm, 165fs, 10MHz, 51.8 mW, polarization as shown in (a), exposure time of 4s. Was an exposure time of 4 seconds used to laser writing the structure? Then it should also be a structure of type 3, not 2, as indicated in the text.

4. The caption of Figure 3 indicates continuum laser 405 nm. Most likely it is a continuous-wave (CW) laser.

Author Response

Dear Reviewer,

Thank you for your remarks. We apologize to have forgotten these corrections.

Best regards.